# Inhibition of Neuromuscular Contractions of Human and Rat Colon by Bergamot Essential Oil and Linalool: Evidence to Support a Therapeutic Action

**DOI:** 10.3390/nu12051381

**Published:** 2020-05-12

**Authors:** Marilisa Straface, Raj Makwana, Alexandra Palmer, Laura Rombolà, Joanne Chin Aleong, Luigi Antonio Morrone, Gareth J. Sanger

**Affiliations:** 1Preclinical and Translational Pharmacology, Department of Pharmacy, Health Science and Nutrition, University of Calabria, 87036 Rende, Italy; m.straface@qmul.ac.uk (M.S.); laura.rombola@unical.it (L.R.); luigi.morrone@unical.it (L.A.M.); 2Blizard Institute, Barts and The London School of Medicine and Dentistry, Queen Mary University of London, London E1 4NS, UK; r.makwana@qmul.ac.uk (R.M.); a.palmer@qmul.ac.uk (A.P.); 3Department of Pathology, Royal London Hospital, Barts Health NHS Trust, London E1 1BB, UK; joanne.chinaleong@nhs.net

**Keywords:** bergamot essential oil, linalool, limonene, human, rat, colon, spasmolytic

## Abstract

Bergamot essential oil (BEO) added to food and drink promotes a citrus flavour. Folklore suggests benefits on gastrointestinal functions but with little supporting evidence. BEO and major constituents (linalool, limonene, linalyl acetate) were therefore examined for any ability to influence neuromuscular contractions of human and rat colon. Circular muscle strips (macroscopically-normal human colon obtained following ethical approval at cancer surgery; Sprague–Dawley rats) were suspended in baths (Krebs solution; 37 °C; 5% CO_2_ in O_2_) for measurement of neuronally-mediated contractions (prevented by tetrodotoxin or atropine) evoked by electrical field stimulation (5 Hz, 0.5 ms pulse width, 10s/minute, maximally-effective voltage), or contractions evoked by KCl (submaximally-effective concentrations). BEO and each constituent concentration dependently inhibited neuronally-mediated and KCl-induced contractions. In human: apparent *p*IC_50_ for BEO (volume/volume Krebs), respectively, 3.8 ± 0.3 and 4.4 ± 0.3; I_max_ 55.8% ± 4.2% and 37.5% ± 4.2%. For the constituents, the rank order of potency differed in human (linalool > limonene >> linalyl-acetate) and rat colon (linalyl-acetate > limonene = linalool), but rank order of efficacy was similar (linalool >> (BEO) = linalyl-acetate >> limonene). Thus, linalool had high efficacy but greater potency in human colon (I_max_ 76.8% ± 6.9%; *p*IC_50_ 6.7 ± 0.2; *n* = 4) compared with rat colon (I_max_ 75.3% ± 1.9%; *p*IC_50_ 5.8 ± 0.1; *n* = 4). The ability of BEO and linalool to inhibit human colon neuromuscular contractility provides a mechanism for use as complementary treatments of intestinal disorders.

## 1. Introduction

Bergamot (*Citrus bergamia Risso et Poiteau)* is a citrus fruit plant growing almost exclusively along the southern coast in the area of Reggio Calabria, Italy (Regional Law 14 Oct 2002, n. 41). Since 2001, the fruit has been given an official designation of “Protected Designation of Origin (DPO)” from the European Union (European regulation CE n. 509/2001). The citrus has been primarily cultivated for its essential oil, bergamot essential oil (BEO), a product in great demand by perfumery and cosmetic industries. BEO or its constituents are also widely used by pharmaceutical, food and confectionery industries as citrus-flavouring agents in food and drink [1].

BEO is obtained by cold pressing of the epicarp and, partly, the mesocarp of the fresh fruit of bergamot (Farmacopea Ufficiale Italiana, 1991). It contains several chemical classes of substances that can be broadly separated into volatile and non-volatile fractions. The former represents 93–96% of total volume of BEO and consists of terpens (e.g., d-limonene, β-bisabolene, γ-terpinene, α-pinene, β-pinene, sabinene, β-myrcene, terpinolene, geranyl acetate) and oxygenated molecules (e.g., linalool, linalyl acetate, neral, geranial, neryl acetate, geranyl acetate) [2,3]. The residual non-volatile fraction (4–7% of total) is comprised of coumarins and psoralens such as bergamottin and citropten [2,3]. Although a large variability exists in the amount and ratio of the individual chemical constituents of the plant, depending on botanical and cultivation factors, three monoterpenes, i.e., limonene, linalyl acetate and linalool, are present in the largest amounts in the BEO, usually 26–53%, 16–40%, and 2–20% by total volume, respectively [4].

The European Medicines Agency (EMA) has documented the anecdotal use of BEO [1], which includes the facilitation of wound healing and its use as an antiseptic, anthelminthic, antimicrobial [5] and antifungal [6] agent. BEO has also been shown to increase oxidative metabolism in human polymorphonuclear leukocytes [7] and affect certain neuronal functions [8]. The latter includes studies, for example, in which low concentrations of BEO were found to increase exocytosis of excitatory amino acids in rat hippocampus, with higher concentrations stimulating the release of glutamate through Ca^2+^-dependent and -independent carriers; the mechanisms of action are unknown [9]. Further, systemic administration of BEO reduced, in a dose-dependent manner, the brain damage caused by focal cerebral ischemia in rats, reducing the levels of the excitatory amino acid glutamate in the penumbral region without influencing the basal amino acid levels [10]. Finally, Rombolà et al. [11] demonstrated anxiolytic-like effects of BEO in rats, comparable with diazepam, but without sedative-like activity.

Food and confectionery industries widely use BEO to promote a citrus flavour in food and drink [1]. However, there is little direct evidence to indicate whether-or-not these substances can affect gastrointestinal (GI) functions and, in particular, influence the neuromuscular activities which govern the movements of the intestine. The possibility that such an activity might occur is supported indirectly by the ability of BEO to influence neuronal functions elsewhere (see above) and also to directly relax the smooth muscle in vascular preparations [12]. In one other study, linalool extracted from elsewhere was found to inhibit contractions evoked by acetylcholine in rat isolated duodenum and ileum [13]. Thus, in the present study, it was hypothesised that BEO and perhaps certain of its major constituents might inhibit GI neuromuscular contractility. This was examined for the first time by using human and rat isolated colon. Previously, human isolated GI tissues have been used to study the effects of several drugs which act directly on the GI tract to promote or inhibit GI motility, with the results having a degree of “translational value”, in that changes in neuromuscular function correlated with their activity in vivo [14].

A preliminary account of some of the data has previously been communicated to the European Society of Neurogastroenterology and Motility [15].

## 2. Methods

### 2.1. Human Colon

Following ethical approval (REC 15/LO/2127), informed written consent was obtained for the use of macroscopically normal ascending and descending/sigmoid (referred to hereon as descending) colon obtained from patients (*n* = 8 females and *n* = 7 males, median age 61 years, ranging between 43–81 years) undergoing elective surgery for non-obstructing bowel cancer and 5–10 cm cut from the tumour. No patient had previous chemoradiotherapy or a diagnosis of inflammatory bowel disease. Tissue was immersed into Krebs solution (in mM: NaCl 118.3, KCl 4.7, MgSO_4_ 1.2, KH_2_PO_4_ 1.2, NaHCO_3_ 25, D-glucose 11.1, CaCl_2_ 2.5), pre-gassed with 95% O_2_ and 5% CO_2_, and within 2 hours after surgery was transferred to the laboratory. The mucosa, muscularis mucosa and submucosal plexus were removed by blunt dissection and discarded. Muscle strips (~15 mm long and ~5 mm wide) were cut approximately parallel to the circular muscle fibres. These were used immediately or after overnight storage at 4 °C in fresh, pre-oxygenated Krebs solution.

### 2.2. Rat Colon

Experiments were carried out in accordance with the directives of the UK Animals (Scientific Procedures) Act 1986 and approved by the animal welfare ethical review board for Queen Mary University of London. All efforts were made to minimise animal suffering and to use only the number of animals necessary to produce reliable results. Male (*n* = 8) and female (*n* = 8) Sprague–Dawley rats weighing 150 g (Charles River Laboratories, Margate, UK) were kept in a controlled temperature (22 ± 1 °C), humidity (55% ± 10%) and 12-hour light–dark cycle. Both sexes were segregated in separate cages with food and water provided ad libitum and were used after a minimum of 6 days of acclimatisation to their new environment following their arrival in the animal unit.

At the beginning of each experiment, the animal was killed by CO_2_ asphyxiation and the colon rapidly resected and immersed in pre-oxygenated (95% O_2_ and 5% CO_2_) Krebs solution. The tissue was cleaned of its intraluminal content, cut into opened rings (about 15 mm long and ~5 mm wide) and used immediately. Although exposure to CO_2_ may adversely affect intestinal motility [16], in the present study, the strips of colon were allowed to recover in an oxygenated, buffered solution (Krebs) before initiating experiments; most strips responded well to EFS and KCl, and only these were used. 

### 2.3. Tissue Bath Technique

The methods have been described in detail elsewhere [17,18]. In brief, each human or rat colon strip was suspended between two platinum wire electrodes (15 mm in length, 10 mm apart) in 10 ml tissue baths containing Krebs solution oxygenated with 95% O_2_ and 5% CO_2_ and maintained at 37 °C. Changes in muscle tension were recorded using an isometric transducer in milli-Newtons (mN) (MLT201/D, AD Instruments, Chalgrove, United Kingdom) connected to an AcqKnowledge data acquisition system version 3.8.1 (BIOPAC Systems Inc., Goleta, CA, USA) on a personal computer (Dell, UK, www.dell.com/uk). The stimulation electrodes were connected to an STG2008 stimulator (Multi Chanel Systems, Reukingen, Germany). The rat and human muscle strips were placed under an initial load of, respectively, 10 and 20 mN, and allowed to equilibrate for 60 min (fresh tissues) or up to 150 min (for those human tissues which were stored overnight), with renewals of the Krebs solution every 15 min.

Electrical field stimulation (EFS) was applied with pulse trains of 5 Hz, pulse width 0.5 ms, for 10 s every 1 min at a voltage 10% higher than that required to obtain maximal contractions [17]. Tissues were allowed to equilibrate for at least 30 (rat) or 60 (human) min, during which the basal muscle tension and the amplitude of the phasic contractions evoked by EFS become stable. Cumulative concentration-response curves were then obtained for BEO (10^−6^ to 10^−3^ % *v/v*), (-)-linalool (10^−9^ to 10^−4^ M), linalyl acetate (10^−9^ to 10^−4^ M) and (R)-(+)-limonene (10^−9^ to 10^−4^ M), with intervals of 15 min between addition of each concentration. For linalool and limonene, the choice of isomers aligned with the % of prevalence in BEO; ~99.5% was (−)-linalool and ~98% was (+)-limonene [2].

Changes in contraction amplitude caused by the addition of BEO or its constituents were quantified as a percentage, by calculating the maximum reduction in the amplitude of the contractions after each addition of the substance (determined as a mean of three consecutive contractions) and comparing with the mean amplitude of three contractions recorded immediately before its first addition. To confirm the neurogenic and cholinergic nature of contractions, some tissues were treated with, respectively, tetrodotoxin (10^−6^ M) or atropine (10^−6^ M).

In other experiments, BEO and its constituents were examined for their ability to relax colon strips pre-contracted with a submaximally-effective concentration of potassium chloride (KCl). Preliminary studies with cumulative additions of KCl (20 to 120 mM, with 15 min intervals between each concentration) identified the concentrations which elicited a 50% of maximal contraction (EC_50_ value) as 72.1 ± 1.1 mM (*n* = 4) and 67.9 ± 0.9 mM (*n* = 3), respectively, in human (ascending and descending) and rat (proximal and distal) colon. A submaximally-effective single concentration of KCl was then selected (40–60 mM) to evoke a sustained tonic contraction ~ 40% of the maximum contraction in both species, prior to constructing a cumulative concentration response curve for BEO (10^−5^ to 10^−3^ % *v/v*) or linalool (10^−7^ to 10^−4^ M) at 15 min dosing intervals. To quantify the change, the muscle tension was measured over at least 5 min prior to the application of BEO or linalool and then again over 5 min after the addition of each concentration, once the response to that concentration had stabilised; these were expressed as mean percentage changes.

In human colon from females and males, the amplitude of the phasic EFS-evoked contractions was, respectively, 8.5 ± 2.1 (*n* = 4) and 7.8 ± 1.8 mN (*n* = 4) whereas the maximum sustained increase in muscle tone caused by the submaximally-effective concentration of KCl was 22.9 ± 7.4 mN (*n* = 4) and 26.4 ± 7.5 (*n* = 3) (*p* > 0.05 each); the former representing approximately 33% of the latter. Similarly, in female and male rats, the amplitude of the phasic EFS-contractions was, respectively, 7.4 ± 1.1 and 6.3 ± 1.2 mN (*n* = 4 each), and the sustained increase in tone generated by KCl was 6.6 ± 1.2 and 5.7 ± 0.6 mN (*n* = 4 each; *p* > 0.05 each), the former representing approximately 111% of the latter.

### 2.4. Materials

(−)-linalool, (R)-(+)-limonene, tetrodotoxin (TTX) and atropine were purchased from Sigma-Aldrich, Poole, UK; linalyl acetate was purchased from Thermo Fisher Scientific, Loughborough, UK. The “whole” BEO sample was used, as it represents the form marketed for personal human use and for therapeutic use. BEO was kindly provided by “Capua Company1880 S.r.l.,” Campo Calabro, Reggio Calabria (Italy) and chromatographic results on the certificate of analysis confirmed that the essential oil contained (R)-(+)-limonene, 48.61%; linalyl acetate, 23.60%; (−)-linalool, 5.52%. BEO, linalool, linalyl acetate and limonene were dissolved in dimethylsulphoxide (DMSO; stock solution of 10^−1^ M). TTX and atropine were dissolved respectively in distilled water and ethanol (stock solution of 10^−3^ M). The total volume of the solvents added to the organ baths did not exceed 1% of the bath volume.

### 2.5. Data Analysis

The cumulative concentration-effect curves in the absence and presence of a test substance were fitted by non-linear regression to a four-parameter Hill equation, and the statistical significance of any difference between unpaired data was determined by Student’s two-tailed *t*-test, using GraphPad PRISM 7.0 for Windows (Graph-Pad Software, La Jolla, CA, USA). The concentration-response data were plotted as the mean ± standard error of the mean (mean ± sem); *n* values are numbers of patients or animals. *p* < 0.05 was considered statistically significant.

The *p*IC_50_ shows the negative logarithm of the concentration of a drug that gave a half-maximal inhibitory response; the I_max_ shows the maximum inhibition obtained. Due to the difficulty of obtaining high concentrations of ligands in solution, it was not always possible to obtain a true maximum of activity. For this reason, apparent *p*IC_50_ and I_max_ values are given.

## 3. Results

### 3.1. Neuronally-Mediated Contractions

In human colon, EFS evoked a marked “after-contraction”, which occurred immediately after termination of EFS; the increase in muscle tension generated at the peak of this contraction was 8.1 ± 1.3 mN (*n* = 8). In 3 of the 8 experiments, EFS also evoked a small contraction during stimulation (1.1 ± 0.1 mN); however, only the after-contraction was used for analysis since it was found in all samples (*n* = 8). In the rat colon, EFS evoked monophasic contractions; the increase in muscle tension generated at the peak of the contraction was 6.9 ± 0.9 mN (*n* = 8). All responses to EFS were prevented by application of TTX 10^−6^ M (human: after-contractions inhibited by 97.9% ± 0.5%, *n* = 4; rat: contractions inhibited by 96.6% ± 4.2%, *n* = 4), consistent with the neurogenic origin of the contractions (Figure 1). Similarly, the contractions were prevented by application of atropine 10^−6^ M (human: after-contractions inhibited by 98.6% ± 0.3%, *n* = 4; rat: contractions inhibited by 97.2% ± 1.7%, *n* = 4), consistent with the predominantly cholinergic nature of the contractions (Figure 1).

Time-matched vehicle controls (Figure 2) showed no statistically significant changes in the amplitude of EFS-evoked contractions between the beginning and end of the experiment (human: 3.9% ± 2.7% inhibition, *n* = 4; rat: 5.9% ± 2.7% inhibition, *n* = 3; *p* > 0.05 each). Application of BEO to the human colon reduced the amplitude of EFS-evoked contractions in a concentration-dependent manner, the activity being consistently observed at 10^−5^ to 10^−3^ M. In these experiments, the results obtained using ascending and descending colon were combined (Figure 2). However, similar activity was observed when using either of these regions separately (Table 1). Overall, the apparent *p*IC_50_ (volume/ volume bathing solution) was 3.8 ± 0.3, and at the highest concentration tested, the contractions were inhibited by 55.8% ± 4.2%.

Application of the different constituents of BEO also caused inhibition of EFS-evoked contractions in human colon (Figure 2 and Table 1). The rank order of potency (apparent *p*IC_50_) was linalool (6.7 ± 0.2) > limonene (5.5 ± 0.2) >> linalyl acetate-ac (4.4 ± 0.4). However, in terms of efficacy (maximum % inhibition or apparent I_max_), the rank order was linalool (76.8% ± 6.9%) >> BEO (55.8% ± 4.2%) = linalyl acetate (53.3% ± 2.9%) >> limonene (27.5% ± 4.3%). Together, these data highlighted linalool as the most efficacious and potent constituent of BEO.

In the rat colon, application of BEO and its constituents reduced the amplitude of EFS-evoked contractions in a concentration-dependent manner. Because similar activity was observed when using either proximal or distal colon (*p* > 0.05), the results obtained using both regions were combined. In terms of potency the rank-order (apparent *p*IC_50_) was linalyl acetate (7 ± 0.2) > limonene (6.1 ± 0.3) = linalool (5.8 ± 0.1). However, in terms of efficacy (apparent I_max_), the rank order was linalool (75.3% ± 1.9%) >> BEO (56.3% ± 2.2%) = linalyl acetate (49.5% ± 1.7%) >> limonene (24.7% ± 1.5%) (Figure 3 and Table 1).

### 3.2. Muscle Contractions Evoked by KCl

Unlike EFS, the submaximally-effective concentration of KCl evoked a short-lived series of phasic contractions followed by loss of these contractions (in all preparations except the rat proximal colon) and the development of a tonic muscle contraction sustained throughout the duration of the experiment (Figure 4 and Figure 5); in the presence of TTX, 1 µM the initial phasic contractions of human colon were absent and KCl only induced a tonic contraction (data not shown). Time-matched exposure to the vehicle did not cause a statistically significant change in the sustained contraction in either human and rat colon Thus, comparing the contraction between the beginning and at the end of the experiment, showed no statistically significant difference (human: 3.2% ± 3.3% inhibition, *n* = 4; rat: 0.9% ± 5.1% inhibition, *n* = 4; Figure 4). However, in colon from both human and rat, cumulative application of BEO or linalool caused a concentration-dependent inhibition of the contraction. In the human colon, the maximum % inhibition of the KCl-evoked contraction (apparent I_max_) caused by BEO was 37.5% ± 4.2%, with an apparent *p*IC_50_ of 4.4 ± 0.3 (Figure 4 for combined results and Table 2 for the singular results obtained in each region of colon). Notably, at the concentrations tested, linalool appeared to more effectively inhibit the contraction when compared with BEO, exhibiting an apparent I_max_ of 53.8% ± 4.6%, *p* < 0.0001, and an apparent *p*IC_50_ of 5.6 ± 0.4 (Figure 4 and Table 2). Similar activity was found in ascending and descending human colon.

In the rat colon, application of BEO also reduced the KCl-evoked contraction with an apparent I_max_ of 26.3% ± 3.8% and apparent *p*IC_50_ of 4.1 ± 0.5 (Figure 5). Compared with BEO, the activity of linalool appeared more efficacious, with an overall apparent I_max_ = 36.1% ± 4.8%, *p* = 0.04 and apparent *p*IC_50_ = 5.4 ± 0.3 (Figure 5 and Table 2). Notably, however, this effect of linalool was greater in the proximal colon (apparent I_max_ 47.6% ± 2.3% and apparent *p*IC_50_ was 5.6 ± 0.2; *n* = 4), compared with the distal colon (apparent I_max_ 21.7% ± 1% and apparent *p*IC_50_ 4.8 ± 0.2; *n* = 4; *p* < 0.05). No statistical difference was observed in the apparent I_max_ and *p*IC_50_ of BEO between proximal and distal colon (*p* > 0.05).

## 4. Discussion

The present study demonstrates, for the first time, the ability of BEO and more, especially linalool, a constituent of BEO, to inhibit cholinergically-mediated contractions of human colon evoked by EFS. Broadly similar activity was also observed when using EFS to evoke cholinergically-mediated contractions of rat isolated colon. Nevertheless, small differences suggested species differences in mechanisms of action. Thus, in the human colon, linalool was the most potent (apparent *p*IC_50_ 6.7) and effective (maximum of 76.8% inhibition) inhibitor of neuromuscular function, and whilst linalool remained the most efficacious inhibitor of neuromuscular function in the rat colon (maximum of 75.3% inhibition), the potency of linalyl acetate (apparent *p*IC_50_ 7; maximum inhibition of 49.5 %) was greater (linalool apparent *p*IC_50_ 5.8).

The application of BEO or linalool also relaxed the muscle of human and rat colon when contracted in a sustained manner by KCl added to the bathing solution. Such a procedure is normally considered to cause contraction of GI preparations largely by depolarization of the smooth muscle with the consequent opening of L-type calcium channels and a rise in intracellular (Ca^2+^) leading to muscle contraction [19,20]. However, a limitation of the present study is that KCl will also depolarize the intrinsic neurons of the colon. For this reason, BEO and linalool were added only after fade of the initial phasic contractions following the application of KCl and replacement by the tonic muscle contraction that was sustained over 60–90 min. Thus, the ability of BEO and linalool (when tested separately) to inhibit the contraction induced by KCl suggests a direct action on the smooth muscle cells. In this activity, the potency of BEO and its constituents appeared to be less than that measured when tested against the EFS-evoked contractions. However, it is difficult to know with any certainty if such differences reflect the differences in types of assay (e.g., measurement of phasic versus tonic contraction) or if this indicates an additional ability of these substances to directly inhibit neuronal functions. Further experiments in both species are needed to examine the actions of BEO and linalool on KCl-induced contractions in the presence of TTX.

Linalool is a constituent of essential oils that are often found in plants with characteristic aroma and flavour [21]. When isolated, linalool has been shown to cause smooth muscle relaxation in animal tissues, including guinea-pig ileum (from lavender; *Lavandula angustifolia* P. Miller) [22] and rat intestine (from South American chemotypes of *Lippia alba*) [13]. In addition, BEO has been shown to cause vasorelaxation of mouse aorta [12], inducing hyperpolarization by activation of potassium channels, an effect partially inhibited by the K^+^ channel blocker tetraethylammonium chloride. In these experiments, the CaCl_2_-induced contraction of the mouse aorta was also inhibited by pre-treatment with BEO, highlighting a direct or indirect ability to block cell membrane calcium channel functions [12]. Thus, from the literature, it may be speculated that BEO or linalool (when tested separately) may inhibit muscle contractility in rat and human colon by acting on voltage-gated calcium channels present on myocyte membranes, reducing intracellular availability of Ca^2+^. Nevertheless, regardless of the mechanism of action at the smooth muscle, an additional ability to directly inhibit neuronal function cannot be excluded.

In different animal models of non-GI activity, BEO has been shown to modulate synaptic function in both physiological and pathological conditions [9,10,11]. Anti-nociceptive activities of BEO and linalool (isolated from various sources) have also been well documented [23]. Vatanparast et al. [24] suggested that the inhibitory action of linalool (10^−4^ M) in central neurons of *Caucotachea atrolabiata* was exerted mainly via voltage-gated sodium channels, reducing the probability of generating an action potential [24]. Another study, which investigated the effects of linalool on various voltage-gated currents in rat sensory neurons, demonstrated that linalool (3 × 10^−3^ M) was a more effective inhibitor of Na^+^ current (IC_50_ 5.6 × 10^−4^ M) compared with other currents [25]. Thus, in summary, we can understand that BEO, acting mainly through the actions of linalool, could act at smooth muscle and/ or neuronal cells by influencing Na^+^, Ca^2+^ and/ or K^+^ intracellular levels [12,24,25]. In muscle cells, linalool may block the membrane voltage-gated calcium channels, reducing intracellular [Ca^2+^] and causing muscle relaxation by reducing the interaction between actin and myosin filaments [26]. In nerve cells, an action of linalool on voltage-dependent sodium channels [24,25] would reduce the probability of generating action potentials, preventing the increase in intracellular Ca^2+^ required to guarantee fusion of synaptic vesicles on the presynaptic membrane, resulting in inhibition of the release of neurotransmitters, including acetylcholine. 

It should be noted that linalool (IC_50_ 1.41 × 10^−4^ M) has also been shown to block 5-HT_3_ receptor activity (a ligand-gated ion channel) in a non-competitive, non-surmountable manner [27]. Further, high concentrations activate the human (10^−4^ M) or mouse (6.7 ± 2.0 × 10^−3^ M) TRPM8 receptor [28,29] and induce a transient increase in short circuit current in mouse isolated duodenum but not colon mucosa (10^−4^ and 3 × 10^−4^ M) [30].

## 5. Conclusions

The results obtained confirm the hypothesis that BEO, largely through the actions of linalool, can inhibit cholinergically-mediated contractions of the human colon, at least partly by acting directly to relax smooth muscle contractility. This activity of BEO must be due to a combination of the effects of the different constituent, but of these, it must be noted that although likely to be present in small amounts, linalool was the most efficacious and most potent. The ability of BEO and its constituents to inhibit neuronally-mediated contractions were similar in the rat colon, although compared with the human colon, the constituents exhibited differences in potency. 

Consequently, the present study supports the use of BEO to promote “gastrointestinal health”, demonstrating a mechanism by which BEO may have potential benefits in the complementary treatment of intestinal diseases related to increased muscle movement. In such patients, muscle relaxants such as the L-type calcium channel blocker pinavarium are, for example, prescribed for treatment of irritable bowel syndrome [31]. Clearly, further studies are needed to clarify the still poorly understood mechanisms of action of BEO and, in particular, demonstrate translation of the present findings by studies in vivo, looking for potential spasmolytic activity in addition to the necessary safety studies. 

## Ackonwledgement

We thank the members of the Barts Health NHS Trust for their contributions in providing human tissue.

## Figures and Tables

**Figure 1 nutrients-12-01381-f001:**
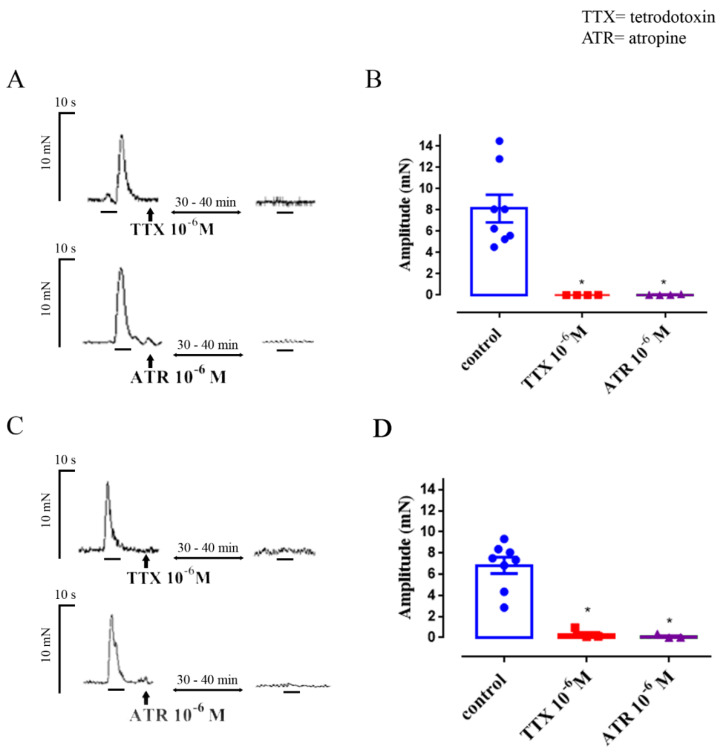
Inhibition by tetrodotoxin and atropine of EFS-evoked contractions of human and rat colon. The representative contraction evoked by EFS before and after administration of tetrodotoxin (TTX) or ATR is showed in panels (**A**) (human) and (**C**) (rat). Panels (**B**) (human) and (**D**) (rat) show the changes in the amplitude of EFS-contractions. Each point represents the mean of patients and animals studied: control *n* = 8, TTX *n* = 4, ATR *n* = 4 in human; control *n* = 8, TTX *n* = 4, ATR *n* = 4 in rat. Vertical lines show standard error of mean. EFS (frequency 5 Hz, pulse width 0.5 ms, for 10 s every 1 min). EFS = electrical field stimulation. * *p* ≤ 0.05 versus control (*t-*tests).

**Figure 2 nutrients-12-01381-f002:**
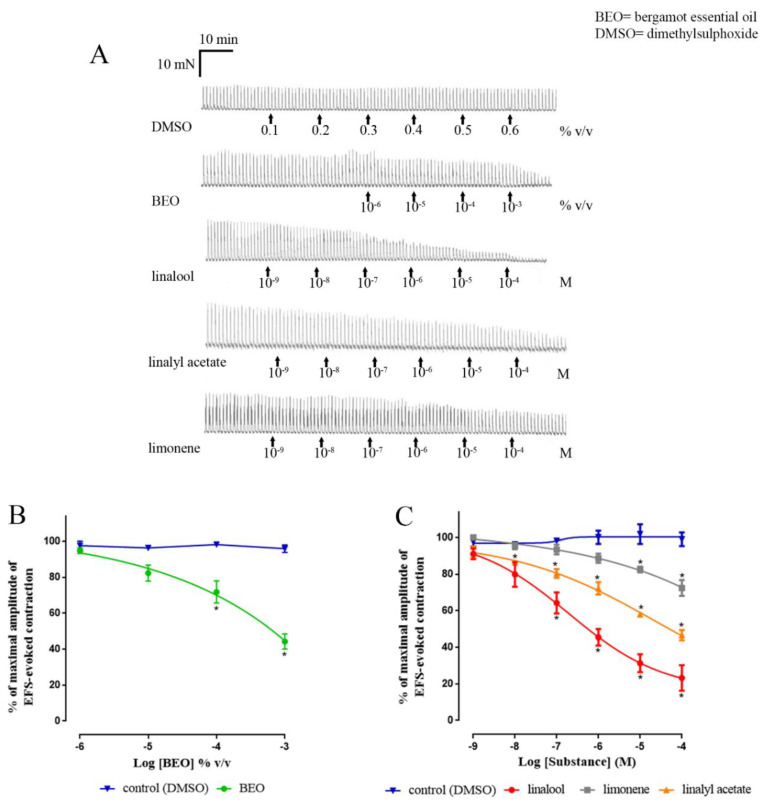
Effect of bergamot essential oil (BEO) and its major components on EFS-evoked contractions in human isolated colon. The examples of experimental records illustrating the pharmacology of responses are showed in panel (**A**). Panel (**B**) shows the concentration–response curves determined by BEO, and panel (**C**) shows the combination of all results obtained with linalool, linalyl acetate, limonene. Note that the values for BEO, an oil, are given as volume/volume of bathing solution, whereas the concentrations of the individual compounds are given as molar values. Each point represents the mean of patients studied: DMSO *n* = 4, BEO *n* = 5, linalool *n* = 4, linalyl acetate *n* = 4, limonene *n* = 3. The cumulative concentration-effect was determined by combining data obtained from both regions of colon (ascending + descending) and were fitted by non-linear regression to a four-parameter Hill equation. Vertical lines show standard error of mean. EFS (5 Hz, pulse width 0.5 ms, for 10 s every 1 min). EFS = electrical field Ssimulation. * *p* ≤ 0.05 versus control (*t-*tests).

**Figure 3 nutrients-12-01381-f003:**
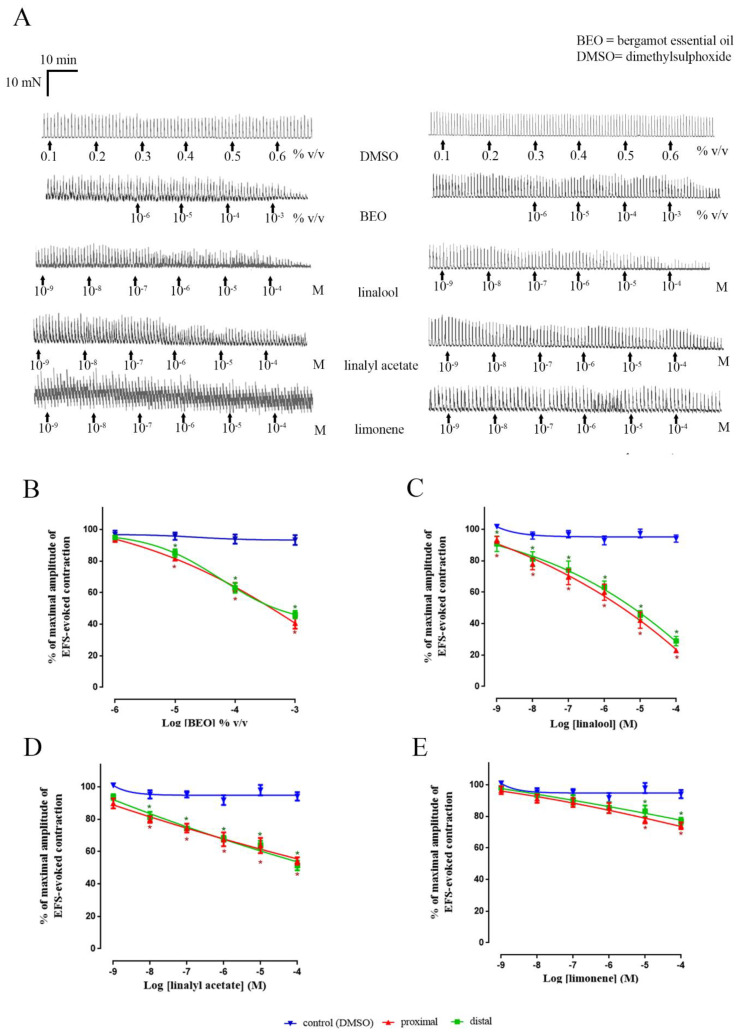
Effect of BEO and its main components on EFS-evoked contractions in rat isolated colon. The examples of experimental records illustrating the pharmacology of responses are shown in panel (**A**) (records on left and right show the effect observed in proximal and distal colon, respectively). Panels (**B–E**) show the concentration–response curves determined by BEO, linalool, linalyl acetate, and limonene in both regions of colon. Note that the values for BEO, an oil, are given as volume/ volume of bathing solution, whereas the concentrations of the individual compounds are given as molar values. Each point represents the mean of animals studied: DMSO *n* = 3, BEO *n* = 4 (*n* = 4 proximal and *n* = 4 distal), linalool *n* = 4 (*n* = 4 proximal and *n* = 4 distal), linalyl acetate *n* = 4 (*n* = 4 proximal and *n* = 4 distal), limonene *n* = 4 (*n* = 4 proximal and *n* = 4 distal). The cumulative concentration-effect were fitted by non-linear regression to a four-parameter Hill equation. Vertical lines show standard error of mean. EFS (5 Hz, pulse width 0.5 ms, for 10 s every 1 min). EFS = electrical field stimulation. * *p* ≤ 0.05 shows the statistical significance between proximal (symbols in red, *) or distal (symbols in green, *) colon versus control (*t-*tests). There was no difference (*p* > 0.05) between the data obtained using the proximal and distal colon.

**Figure 4 nutrients-12-01381-f004:**
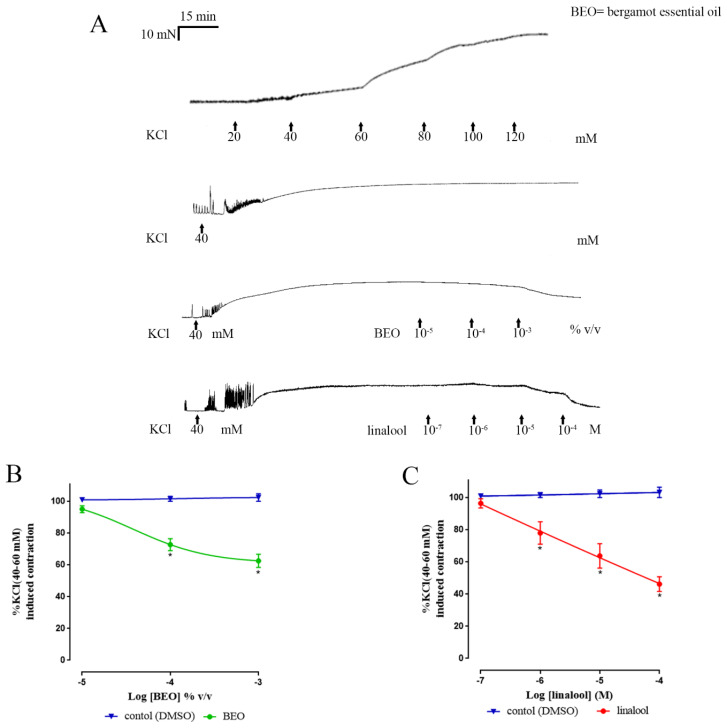
Inhibition by BEO and linalool of the contraction evoked by a submaximally-effective concentration of KCl in human isolated colon. Panel (**A**) shows examples of experimental records illustrating a cumulative concentration-response curve to KCl and the effects of vehicle, BEO and linalool on the sustained contraction evoked by the submaximally-effective concentration of KCl. Panel (**B**) shows the concentration–response curve for BEO and panel (**C**) shows the concentration–response curve for linalool. Note that the values for BEO are given as volume/ volume of bathing solution, whereas the concentrations of linalool are given as molar values. Each point represents the mean of patients studied: BEO *n* = 4 and linalool *n* = 4. The cumulative concentration-effect was determined by combined data obtained of both regions of colon (ascending + descending) and were fitted by non-linear regression to a three or four-parameter Hill equation. Vertical lines show mean ± sem. * *p* ≤ 0.05 versus control (*t-*tests).

**Figure 5 nutrients-12-01381-f005:**
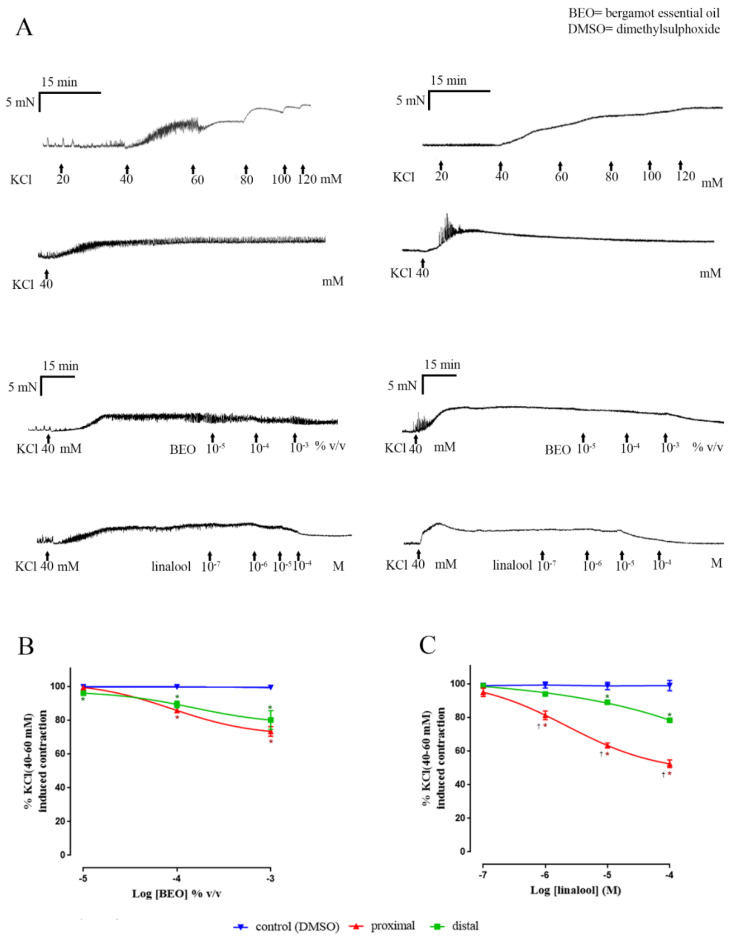
Inhibition by BEO and linalool of the contraction evoked by a submaximally-effective concentration of KCl in rat isolated colon. Panel (**A**) shows examples of experimental records (records on left and right show the effect observed in proximal and distal colon, respectively) illustrating a cumulative concentration-response curve to KCl and the effects of vehicle, BEO and linalool on the sustained contraction evoked by the submaximally-effective concentration of KCl. Panel (**B**) shows the concentration–response curve for BEO and panel (**C**) shows the concentration–response curve for linalool in both regions of colon. Note that the values for BEO are given as volume/ volume of bathing solution, whereas the concentrations of linalool are given as molar values. Each point represents the mean of animals studied: BEO *n* = 4 (*n* = 4 proximal and *n* = 4 distal) and linalool *n* = 4 (*n* = 4 proximal and *n* = 4 distal). The cumulative concentration effect was fitted by non-linear regression to a three or four-parameter Hill equation. Vertical lines show mean ± sem. * *p* ≤ 0.05 shows the statistical significance between proximal (symbols in red, *) or distal (symbols in green, *) colon versus control (*t-*tests). † *p* < 0.05 shows the statistical significance between the inhibition produced in proximal and distal colon (*t*-tests).

**Table 1 nutrients-12-01381-t001:** Apparent *p*IC_50_ and apparent I_max_ for the inhibition of EFS-evoked contractions in human and rat colon.

**Human**									
	**Ascending**			**Descending**			**Ascending + Descending**		
	**Apparent *p*IC_50_**	**Apparent I_max_ (%**)	***n***	**Apparent *p*IC_50_**	**Apparent I_max_ (%**)	***n***	**Apparent *p*IC_50_**	**Apparent I_max_ (%**)	***n***
Control (DMSO)	-	4.6 ± 1.4	2	-	3.2 ± 3.2	2	-	3.9 ± 2.7	4
BEO	4.7 ± 0.3	59.1 ± 6.6	2	3.1 ± 0.6	53.1 ± 11.3	3	3.8 ± 0.3	55.8 ± 4.2 *	5
linalool	6.3 ± 0.3	88.8 ± 5.4	2	6.8 ± 0.2	64.8 ± 3.2	2	6.7 ± 0.2	76.8 ± 6.9 *	4
linalyl acetate	4.7 ± 0.6	58.4 ± 0.4	2	6.3 ± 0.3	48.4 ± 0.2	2	4.4 ± 0.4	53.3 ± 2.9 *	4
limonene	5.2 ± 0.3	30.6 ± 5.3	2	5.9	21.4	1	5.5 ± 0.2	27.5 ± 4.3 *	3
**Rat**									
	**Proximal**			**Distal**			**Proximal + Distal**		
	**Apparent *p*IC50**	**Apparent Imax (%**)	***n***	**Apparent *p*IC50**	**Apparent Imax (%**)	***n***	**Apparent *p*IC50**	**Apparent Imax (%**)	***n***
Control (DMSO)	-	6.8 ± 4.7	3	-	4.9 ± 3.5	2	-	5.9 ± 2.7	5 (3 rats)
BEO	4.2 ± 0.2	60.8 ± 3.49 *	4	4.1 ± 0.1	53.7 ± 2.4 *	4	4 ± 0.3	56.3 ± 2.2 *	8 (4 rats)
linalool	5.8 ± 0.2	77.8 ± 1.3 *	4	5.8 ± 0.2	69.4 ± 3 *	4	5.8 ± 0.1	75.3 ± 1.9 *	8 (4 rats)
linalyl acetate	6.8 ± 0.3	44.5 ± 2.2 *	4	7 ± 0.3	49.7 ± 3 *	4	7 ± 0.2	49.5 ± 1.7 *	8 (4 rats)
limonene	5.9 ± 0.3	25.2 ± 2.1 *	4	6.4 ± 0.4	23.1 ± 2.2 *	4	6.1 ± 0.3	24.7 ± 1.5 *	8 (4 rats)

Comparison of *p*IC_50_ and I_max_ for the inhibitory effect on EFS -mediated contractions in human and rat isolated colon. BEO = bergamot essential oil. Note that the values for BEO are given as volume/ volume of bathing solution, whereas the concentrations of the individual compounds are given as molar values. Data are shown separately for each region of colon and when combined (ascending + descending, proximal + distal) and are given as means and standard errors of the mean. * *p* < 0.05 shows the statistical significance between the highest concentrations of BEO or linalool tested on EFS-contraction versus EFS-contraction control (*t*-tests). For the human colon, statistical tests were carried out only for the combined data and not for the proximal and distal human colon separately as the *n*-values were too small. *N* = number of patients or animals studied.

**Table 2 nutrients-12-01381-t002:** Apparent *p*IC_50_ and apparent I_max_ for the inhibition of KCl-evoked contractions in human and rat colon.

**Human**									
	**Ascending**			**Descending**			**Ascending + Descending**		
	**Apparent *p*IC_50_**	**Apparent I_max_ (%)**	***n***	**Apparent *p*IC_50_**	**Apparent I_max_ (%)**	***n***	**Apparent *p*IC_50_**	**Apparent I_max_ (%)**	***n***
Control (40–60 mM KCl)	-	6.41 ± 6.3	2	-	0.2 ± 0.1	2	-	3.2 ± 3.3	4
BEO	4.4 ± 0.8	38.8 ± 19.6	2	4.5 ± 0.2	36.1 ± 2.9	2	4.4 ± 0.3	37.5 ± 4.2 *	4
linalool	5.3 ± 0.2	58.9 ± 4.3	2	5.9 ± 0.5	52.3 ± 4.2	2	5.6 ± 0.4	53.8 ± 4.6 *	4
**Rat**									
	**Proximal**			**Distal**			**Proximal + Distal**		
	**Apparent *p*IC_50_**	**Apparent I_max_ (%)**	***n***	**Apparent *p*IC_50_**	**Apparent I_max_ (%)**	***n***	**Apparent *p*IC_50_**	**Apparent I_max_ (%)**	***n***
Control (40–60 mM KCl)	-	1.4 ± 0.73	4	-	0.6 ± 5.6	4	-	0.9 ± 5.1	8 (4 rats)
BEO	4.4 ± 0.4	32.8 ± 3.2 *	4	3.7 ± 0.4	19.9 ± 5.5 *	4	4.1 ± 0.5	26.3 ± 3.8 *	8 (4 rats)
linalool	5.6 ± 0.2	47.6 ± 2.3 *^,^^†^	4	4.8 ± 0.2	21.7 ± 1 *	4	5.4 ± 0.3	36.1 ± 4.8 *	8 (4 rats)

Comparison of *p*IC_50_ and I_max_ for the inhibition effect on KCl-mediated contractions in human and rat isolated colon. BEO = bergamot essential oil. Note that the values for BEO are given as volume/volume of bathing solution, whereas the concentrations of the individual compounds are given as molar values. Data are shown separately for each region of colon and when combined (ascending + descending, proximal + distal) and are given as means and standard errors of the mean. *****
*p* < 0.05 shows the statistical significance between the highest concentrations of BEO or linalool tested on KCl-contraction versus KCl-contraction control (*t*-tests); ^†^
*p* < 0.05 shows the statistical significance between the inhibition produced in proximal and distal colon (*t*-tests). For the human colon, statistical tests were carried out only for the combined data and not for the ascending and descending colon separately as the *n*-values were too small. *N* = number of patients and animals studied except when data from rat proximal and distal colon are combined (usually from the same animal), when *N* = number of tissues studied.

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
