# Peer review of "Inhibition of Neuromuscular Contractions of Human and Rat Colon by Bergamot Essential Oil and Linalool: Evidence to Support a Therapeutic Action"

_nutrients, 2020, doi:10.3390/nu12051381_

Round 1

Reviewer 1 Report

In this work Straface et al. study the effects of bergamot essential oil and their components, linalool, limonene and linalyl acetate on the muscular neurotransmission of the human and rat colon. Both bergamot and its components inhibited neuronally-mediated and KCl-induced contractions in the rat and human colon, although the potency and efficacy of these compounds were different in the two species studied.

The study of bergamot as a possible spasmolytic treatment against functional intestinal disorders is interesting.

Although the results and conclusions shown in the study seem consistent, the methodology used in relation to the collection of intestinal samples and the analysis of the data raises some doubts.

Major comments

1. The authors use CO2 to humanely sacrifice animals. Could the authors document with bibliography that the slaughter of animals with CO2 has no effect on spontaneous intestinal motility?

2. The study uses circular muscle strips from the colon of humans and rats of both sexes and it is indicated that there are no differences between males and females. It would be nice if you could indicate in the work the statistical analyzes, including the confidence levels reached (p values), that have been carried out to reach this conclusion.

3. In the case of the rat colon, the authors combine the results obtained in the proximal and distal colon. However, it has been described in the literature that the spontaneous intestinal motility pattern of the proximal colon is different from those of the distal colon. In fact, as seen in Figure 5, the recordings shown with the KCl and BEO incubation appear to correspond to the proximal colon, while the linalool recording appears to correspond to the distal colon. Figure 4 also appears to show recordings of the proximal and distal colon. Therefore, authors should perform independent statistical analyzes for the proximal and distal colon and show the results in the manuscript. All the records represented in the figures should be either proximal colon or distal colon, so that the effects can be compared.

Minor comments

4. In the methodology section, the authors should indicate the number of total rats used in the study.

5. In Figures 4 and 5, authors should indicate in the BEO and linalol records at what time throughout the recording the KCl was added.

Author Response

Referee 1

Major comments

  1. The authors use CO2 to humanely sacrifice animals. Could the authors document with bibliography that the slaughter of animals with CO2 has no effect on spontaneous intestinal motility?

Reply: Thank you for this comment. We agree that changes in blood pH caused by CO2 exposure could affect gastrointestinal motility (a new reference has been inserted) but we have added a sentence to point out that once removed from the animals, muscle strips were equilibrated in oxygenated and buffered Krebs solution to allow for recovery.  Most muscle strips responded to subsequent electrical field stimulation or to application of KCl (spontaneous contractions were not studied).

  1. The study uses circular muscle strips from the colon of humans and rats of both sexes and it is indicated that there are no differences between males and females. It would be nice if you could indicate in the work the statistical analyzes, including the confidence levels reached (p values), that have been carried out to reach this conclusion.

            Reply: We agree with the referee and these data are now inserted in the Methods section.

  1. In the case of the rat colon, the authors combine the results obtained in the proximal and distal colon. However, it has been described in the literature that the spontaneous intestinal motility pattern of the proximal colon is different from those of the distal colon. In fact, as seen in Figure 5, the recordings shown with the KCl and BEO incubation appear to correspond to the proximal colon, while the linalool recording appears to correspond to the distal colon. Figure 4 also appears to show recordings of the proximal and distal colon. Therefore, authors should perform independent statistical analyzes for the proximal and distal colon and show the results in the manuscript. All the records represented in the figures should be either proximal colon or distal colon, so that the effects can be compared.

            Reply: Thank you for this comment. We agree on the possibility of region-dependent differences (spontaneous contractions were not examined; we induced contractions by exposures to standard periods of electrical field stimulation and by exposure to KCl).  The separate data for the two regions have now been inserted within 2 new Tables added to the manuscript, summarising key data for human and rat colon and showing the data for the ascending/ proximal colon and descending/ distal colon separately and together with the combined data.  Statistical analysis is applied to the data from each region of the rat colon but not for the human colon as the numbers were too small. This means that for the human colon the graphs still show the data combined for the two regions whereas the regional and combined data are shown together in the figures which illustrate the rat data.

Minor comments

  1. In the methodology section, the authors should indicate the number of total rats used in the study.

Reply: This has been added.

  1. In Figures 4 and 5, authors should indicate in the BEO and linalol records at what time throughout the recording the KCl was added.

Reply: The records in figures 4 and 5 have been changed to show the time of administration of KCl, BEO or linalool.

Reviewer 2 Report

Manuscript ID: nutrients-783366

Inhibition of neuromuscular contractions of human and rat colon by bergamot essential oil and linalool: evidence to support a therapeutic action.

An interesting study.

BEO may not only be beneficial but in higher dosages also neurotoxic inducing fasciculations

We should know if any side effects were observed

Were different dosages tested?

Author Response

Referee 2

BEO may not only be beneficial but in higher dosages also neurotoxic inducing fasciculations

We should know if any side effects were observed

Were different dosages tested?

Reply: Our study was conducted in vitro and a range of different concentrations were tested. We thank the referee for reminding us of the need for safety studies so in the last sentence of the Discussion we have pointed out the need for further testing of BEO not just to examine efficacy but also safety.

Reviewer 3 Report

The present manuscript submitted by Straface et al. presented the inhibitory effect of bergamot essential oil (BEO) and its major compounds especially linalool on colon contractile activity. Experiments performed on rat and human colon samples showed that BEO and linalool inhibited electrically field stimulated and KCL-induced contractions of colon.

The study is novel and original however serious methodologic flaws and weaknesses are observed.

Major concerns:

1- BEO composition and respective compound efficiencies. As the authors know BEO composition and the respective rates for each compound, they have to evaluate the respective compounds’ concentrations in BEO solution. Indeed, linalool and limonene contents are 5.52% and 48.61% respectively, namely around ten times more limonene than linalool. This ratio (10) is also found for pIC50. Limonene is around 10 time less potent than linalool but in BEO and considering the respective ratios, these two compounds have the same efficiency. This is an important point to take into account. Thus results interpretation and conclusions must be adapted (line 261; 275).

Another comment concerns BEO pIC50. As BEO concentration is expressed as % to what correspond its pIC50 value? What is the scientific relevance of the comparison with the other compounds’ pIC50 values?

2- Lines 229-235 and line 277: the authors explained that the inhibition of KCl-induced contraction likely demonstrated a direct inhibitory effect on smooth muscle cells. This is not correct. Indeed, KCl is a depolarizing agent and depolarizes smooth muscle cells thus opening voltage-gated Ca2+ channels and triggering contraction. But KCl also depolarizes terminal nerve endings thus activating neurogenic system like EFS (opening of voltage-gated sodium channels, triggering neurotransmitter release and smooth muscle contraction). To address this issue, the authors should perform experiments at least in the presence of atropine to block neurogenic control.

Thus, what are the effects of BEO and linalool on KCl-induced contraction in the presence of atropine?

Additionally, to characterize these inhibitory effects, are the contractile responses to EFS and KCl inhibited by BEO and the various compounds? Namely, what are the effects of EFS and KCl in the presence of BEO and linalool?

The authors speculated inhibitory effects on voltage-gated calcium channels (line 263). What are the effects of BEO in the presence of Ca2+ channels blockers like verapamil or nifedipine?

3- In figure 1: the authors explained that in human colon EFS induced a small contraction followed by a marked after-contraction. Is this always observed? This is not illustrated in Fig1A lower trace. Are the inhibitory effects of BEO and linalool observed on both contractions?

Author Response

Referee 3

Major concerns:

1- BEO composition and respective compound efficiencies. As the authors know BEO composition and the respective rates for each compound, they have to evaluate the respective compounds’ concentrations in BEO solution. Indeed, linalool and limonene contents are 5.52% and 48.61% respectively, namely around ten times more limonene than linalool. This ratio (10) is also found for pIC50. Limonene is around 10 time less potent than linalool but in BEO and considering the respective ratios, these two compounds have the same efficiency. This is an important point to take into account. Thus results interpretation and conclusions must be adapted (line 261; 275).

Another comment concerns BEO pIC50. As BEO concentration is expressed as % to what correspond its pIC50 value? What is the scientific relevance of the comparison with the other compounds’ pIC50 values?

            Reply: Thank you for this careful analysis. Our data with BEO reflects the sum of the different efficacies and potencies of its different constituents. To avoid confusion when discussing the actions of BEO and linalool together we have inserted the words ‘(when tested seperately)’, to indicate that the findings with linalool alone cannot necessarily apply to BEO. We also accept the difficulty of trying to generate comparative numbers on concentrations for an oil versus solid compounds.  Nevertheless, we hope the referee agrees that it is important to quantify data.  In the legends for each figure and table we have now stated that the values for BEO, an oil, are given as volume/ volume of bathing solution whereas the concentrations of the individual compounds are given as molar values.

2- Lines 229-235 and line 277: the authors explained that the inhibition of KCl-induced contraction likely demonstrated a direct inhibitory effect on smooth muscle cells. This is not correct. Indeed, KCl is a depolarizing agent and depolarizes smooth muscle cells thus opening voltage-gated Ca2+ channels and triggering contraction. But KCl also depolarizes terminal nerve endings thus activating neurogenic system like EFS (opening of voltage-gated sodium channels, triggering neurotransmitter release and smooth muscle contraction). To address this issue, the authors should perform experiments at least in the presence of atropine to block neurogenic control.

Thus, what are the effects of BEO and linalool on KCl-induced contraction in the presence of atropine?

Additionally, to characterize these inhibitory effects, are the contractile responses to EFS and KCl inhibited by BEO and the various compounds? Namely, what are the effects of EFS and KCl in the presence of BEO and linalool?

The authors speculated inhibitory effects on voltage-gated calcium channels (line 263). What are the effects of BEO in the presence of Ca2+ channels blockers like verapamil or nifedipine?

            Reply: The ability of KCl to depolarise both smooth muscle and nerve cells should have been clear.  We have made the following changes. [1] Methods: the size of the sustained increase in muscle tone caused by KCl has been compared with the amplitude of the phasic EFS-induced contractions. [2] Results: Section 3.2: This now begins with the statements “Unlike EFS, the submaximally-effective concentration of KCl evoked a short-lived series of phasic contractions which were followed by a single tonic muscle contraction sustained throughout the duration of the experiment (Figures 4 and 5); in the presence of TTX 1 µM the initial phasic contractions were absent but KCl only induced a tonic contraction (data not shown).” [3] Discussion:  Paragraph 2 begins with the following: “The application of BEO or linalool also relaxed the muscle of human and rat colon when contracted in a sustained manner by KCl added to the bathing solution. Such a procedure is normally considered to cause contraction of GI preparations largely by depolarization of the smooth muscle with consequent opening of L-type calcium channels and a rise in intracellular [Ca2+] leading to muscle contraction [19, 20].  However, since a small contribution caused by activation of intrinsic neurons cannot be fully excluded, BEO and linalool were added only after fade of the initial phasic contractions following application of KCl and replacement by the tonic muscle contraction that was sustained over 60-90 min.

Finally, we could not conduct experiments with nifedipine or verapamil because L-type calcium channel blockers prevent smooth muscle contraction and hence, it would not be possible to take measurements.

3- In figure 1: the authors explained that in human colon EFS induced a small contraction followed by a marked after-contraction. Is this always observed? This is not illustrated in Fig1A lower trace. Are the inhibitory effects of BEO and linalool observed on both contractions?

            Reply:  From 8 patients, 20 total strips were used for EFS experiments, in which 3/20 strips (of 3 patients) showed a small contraction during EFS followed by a marked. In terms of amplitude, the small contraction was 1.1 ± 0.1 mN and the marked contraction was 8.3 ± 2.9 mN. We decided to not quantify the effects of BEO and linalool on the small contractions, which occurred during EFS, because these were inconsistent in the totality of experiments done.

Round 2

Reviewer 1 Report

The authors have addressed all the comments of the reviewer.

Author Response

No changes needed

Reviewer 3 Report

You will find below your answer and my reply for each previuos comment.

Comment 1:

Author’s Reply: Thank you for this careful analysis. Our data with BEO reflects the sum of the different efficacies and potencies of its different constituents. To avoid confusion when discussing the actions of BEO and linalool together we have inserted the words ‘(when tested seperately)’, to indicate that the findings with linalool alone cannot necessarily apply to BEO.We also accept the difficulty of trying to generate comparative numbers on concentrations for an oil versus solid compounds.  Nevertheless,we hope the referee agrees that it is important to quantify data.  In the legends for each figure and table we have now stated that the values for BEO, an oil, are given as volume/ volume of bathing solution whereas the concentrations of the individual compounds are given as molar values

Reviewer reply:

Our data with BEO reflects the sum of the different efficacies and potencies of its different constituents.

What do the authors mean by this sentence? The effects of BEO are not the sum but a combination of the effects of the different constituents. According to BEO composition which was given in the text (line 161) it is possible to evaluate each constituent ratio in BEO and to discuss about their contribution in BEO effects. As I mentioned in my previous comment, although linalool is ten times more potent (pIC50 10 times higher) than limonene, linalool ratio (5.52%) in BEO is around ten times lower than limonene ratio (48.61%). Thus the two constituents may have the same contribution in BEO effects and the authors cannot conclude that the ability of BEO to inhibit human colon neuromuscular contractility occurs largely via linalool (lines 30, 395). This is not true. One way to definitely address this issue could be to mix the three constituents in the same ratios as that found in BEO and to test the efficiency of the mixture. Otherwise the only conclusion is that when tested separately, linalool is the most potent and efficient.

Moreover, as BEO concentration is expressed in %, it is not pharmacologically correct to compare its pIC50 value to those of other constituents (lines 26 to 28 and 212 to 215).

Comment 2:

 Author’s Reply: The ability of KCl to depolarise both smooth muscle and nerve cells should have been clear. We have made the following changes. [1] Methods: the size of the sustained increase in muscle tone caused by KCl has been compared with the amplitude of the phasic EFS-induced contractions. [2] Results: Section 3.2: This now begins with the statements “Unlike EFS, the submaximally-effective concentration of KCl evoked a short-lived series of phasic contractions which were followed by a single tonic muscle contraction sustained throughout the duration of the experiment (Figures 4 and 5); in the presence of TTX 1 µM the initial phasic contractions were absent but KCl only induced a tonic contraction (data not shown).” [3] Discussion: Paragraph 2 begins with the following: “The application of BEO or linalool also relaxed the muscle of human and rat colon when contracted in a sustained manner by KCl added to the bathing solution. Such a procedure is normally considered to cause contraction of GI preparations largely by depolarization of the smooth muscle with consequent opening of L-type calcium channels and a rise in intracellular [Ca2+] leading to muscle contraction [19, 20]. However, since a small contribution caused by activation of intrinsic neurons cannot be fully excluded, BEO and linalool were added only after fade of the initial phasic contractions following application of KCl and replacement by the tonic muscle contraction that was sustained over 60-90 min.

Finally, we could not conduct experiments with nifedipine or verapamil because L-type calcium channel blockers prevent smooth muscle contraction and hence, it would not be possible to take measurements.

Reviewer reply:

The authors admit that TTX modify KCl-induced contractile response “in the presence of TTX 1 µM the initial phasic contractions were absent but KCl only induced a tonic contraction (data not shown)”. Thus, this demonstrates that KCl not only depolarizes smooth muscle cells but also nerve endings. The authors mentioned that TTX suppressed phasic contractions thus implying that the tonic contraction is not modified. It is not sure that nerve endings are not involved in KCl-induced sustained contraction. How this was validated? Did the authors have compared the amplitude of the sustained contraction in absence and in presence of TTX? The authors just mentioned that TTX supressed the phasic contraction but they also have to confirm that the sustained contraction (Emax) is not affected. I confirm my previous comment: some additional experiments are needed. The effects of BEO and linalool have to be studied on KCl-induced response in the presence of either TTX or atropine. Additionally, EFS and KCl responses have to be studied in the presence of BEO and in the presence of linalool. Do these compounds will inhibit phasic and/or tonic contraction? This is an important point to address and which will improve the discussion.

Moreover, the new figure 5 panel A clearly shows that in in proximal colon (left traces) the firing phasic contractions maintained during the plateau phase of contraction. This demonstrates that in distal and in proximal parts different signalling pathways are involved in KCl response. Thus, the requested experiments in the presence of TTX are mandatory to discriminate neurogenic and smooth muscle regulation.

Comment 3:

Author’s Reply: From 8 patients, 20 total strips were used for EFS experiments, in which 3/20 strips (of 3 patients) showed a small contraction during EFS followed by a marked. In terms of amplitude, the small contraction was 1.1 ± 0.1 mN and the marked contraction was 8.3 ± 2.9 mN. We decided to not quantify the effects of BEO and linalool on the small contractions, which occurred during EFS, because these were inconsistent in the totality of experiments done.

Reviewer reply: OK

Additional comment:

If n=4 is allowed for human colon regarding the difficulty in obtaining human samples, for the rat colon it is very low and for statistical analysis even irrelevant (n≥5 at least). The number of individuals should have been increased.

Author Response

Comment 1:

Author’s Reply: Thank you for this careful analysis. Our data with BEO reflects the sum of the different efficacies and potencies of its different constituents. To avoid confusion when discussing the actions of BEO and linalool together we have inserted the words ‘(when tested seperately)’, to indicate that the findings with linalool alone cannot necessarily apply to BEO.We also accept the difficulty of trying to generate comparative numbers on concentrations for an oil versus solid compounds.  Nevertheless,we hope the referee agrees that it is important to quantify data.  In the legends for each figure and table we have now stated that the values for BEO, an oil, are given as volume/ volume of bathing solution whereas the concentrations of the individual compounds are given as molar values

Reviewer reply:

Our data with BEO reflects the sum of the different efficacies and potencies of its different constituents.

What do the authors mean by this sentence? The effects of BEO are not the sum but a combination of the effects of the different constituents. According to BEO composition which was given in the text (line 161) it is possible to evaluate each constituent ratio in BEO and to discuss about their contribution in BEO effects. As I mentioned in my previous comment, although linalool is ten times more potent (pIC50 10 times higher) than limonene, linalool ratio (5.52%) in BEO is around ten times lower than limonene ratio (48.61%). Thus the two constituents may have the same contribution in BEO effects and the authors cannot conclude that the ability of BEO to inhibit human colon neuromuscular contractility occurs largely via linalool (lines 30, 395). This is not true. One way to definitely address this issue could be to mix the three constituents in the same ratios as that found in BEO and to test the efficiency of the mixture. Otherwise the only conclusion is that when tested separately, linalool is the most potent and efficient.

Moreover, as BEO concentration is expressed in %, it is not pharmacologically correct to compare its pIC50 value to those of other constituents (lines 26 to 28 and 212 to 215).

Our Reply

  1. Our thanks to the reviewer for insisting on these corrections which have undoubtably improved the accuracy of the manuscript and which were not fully dealt with previously.
  2. The summary has been changed to make clear that the potency values for BEO refer to volume/ volume bathing solution and reference to BEO has been deleted from the rank order of potency. Additional small adjustments were made to the Summary in order to comply with the 200 word limit.
  3. Last line of the summary now reads: “The ability of BEO and linalool to inhibit human colon neuromuscular contractility, largely via linalool, provides a mechanism for use as a complementary treatment of intestinal disorders.”
  4. The Results for the human colon have been corrected to (a) remove the statement that the constituents were more potent than BEO, (b) remove BEO from the rank order of potency values of the constituents and (c) make it clear that linalool was the most ‘efficacious and potent’ constituent, not the most ‘active’. Small, similar changes were made in the sections describing the Rat Colon data.
  5. First line of Discussion – with reference to linalool the sentence ‘a major constituent of BEO’ has been changed to ‘a constituent of BEO’,
  6. Second line of the conclusion now reads: “This activity of BEO must be due to a combination of the effects of the different constituent, but of these, it must be noted that although likely to be present in small amounts, linalool was the most efficacious and most potent.”

Comment 2:

 Author’s Reply: The ability of KCl to depolarise both smooth muscle and nerve cells should have been clear. We have made the following changes. [1] Methods: the size of the sustained increase in muscle tone caused by KCl has been compared with the amplitude of the phasic EFS-induced contractions. [2] Results: Section 3.2: This now begins with the statements “Unlike EFS, the submaximally-effective concentration of KCl evoked a short-lived series of phasic contractions which were followed by a single tonic muscle contraction sustained throughout the duration of the experiment (Figures 4 and 5); in the presence of TTX 1 µM the initial phasic contractions were absent but KCl only induced a tonic contraction (data not shown).” [3] Discussion: Paragraph 2 begins with the following: “The application of BEO or linalool also relaxed the muscle of human and rat colon when contracted in a sustained manner by KCl added to the bathing solution. Such a procedure is normally considered to cause contraction of GI preparations largely by depolarization of the smooth muscle with consequent opening of L-type calcium channels and a rise in intracellular [Ca2+] leading to muscle contraction [19, 20]. However, since a small contribution caused by activation of intrinsic neurons cannot be fully excluded, BEO and linalool were added only after fade of the initial phasic contractions following application of KCl and replacement by the tonic muscle contraction that was sustained over 60-90 min.

Finally, we could not conduct experiments with nifedipine or verapamil because L-type calcium channel blockers prevent smooth muscle contraction and hence, it would not be possible to take measurements.

Reviewer reply:

The authors admit that TTX modify KCl-induced contractile response “in the presence of TTX 1 µM the initial phasic contractions were absent but KCl only induced a tonic contraction (data not shown)”. Thus, this demonstrates that KCl not only depolarizes smooth muscle cells but also nerve endings. The authors mentioned that TTX suppressed phasic contractions thus implying that the tonic contraction is not modified. It is not sure that nerve endings are not involved in KCl-induced sustained contraction. How this was validated? Did the authors have compared the amplitude of the sustained contraction in absence and in presence of TTX? The authors just mentioned that TTX supressed the phasic contraction but they also have to confirm that the sustained contraction (Emax) is not affected. I confirm my previous comment: some additional experiments are needed. The effects of BEO and linalool have to be studied on KCl-induced response in the presence of either TTX or atropine. Additionally, EFS and KCl responses have to be studied in the presence of BEO and in the presence of linalool. Do these compounds will inhibit phasic and/or tonic contraction? This is an important point to address and which will improve the discussion.

Moreover, the new figure 5 panel A clearly shows that in in proximal colon (left traces) the firing phasic contractions maintained during the plateau phase of contraction. This demonstrates that in distal and in proximal parts different signalling pathways are involved in KCl response. Thus, the requested experiments in the presence of TTX are mandatory to discriminate neurogenic and smooth muscle regulation.

Our reply

We cannot state with 100% certainty that the tonic contraction evoked by the concentration of KCl used in our experiments will not be influenced in any way by the application of TTX. We must, therefore, conduct additional experiments to examine the effect of TTX on the KCl-induced contraction, in both regions of the human and rat colon. If activity is observed, then the experiments with BEO/ linalool must be repeated.

Comment 3:

Author’s Reply: From 8 patients, 20 total strips were used for EFS experiments, in which 3/20 strips (of 3 patients) showed a small contraction during EFS followed by a marked. In terms of amplitude, the small contraction was 1.1 ± 0.1 mN and the marked contraction was 8.3 ± 2.9 mN. We decided to not quantify the effects of BEO and linalool on the small contractions, which occurred during EFS, because these were inconsistent in the totality of experiments done.

Reviewer reply: OK

Additional comment:

If n=4 is allowed for human colon regarding the difficulty in obtaining human samples, for the rat colon it is very low and for statistical analysis even irrelevant (n≥5 at least). The number of individuals should have been increased.

Our Reply

It is not uncommon to apply statistical tests to data derived from n=4, in which tight standard errors can clearly be seen.